# Machine Assisted Experimentation of Extrusion-Based Bioprinting Systems

**DOI:** 10.3390/mi12070780

**Published:** 2021-06-30

**Authors:** Shuyu Tian, Rory Stevens, Bridget T. McInnes, Nastassja A. Lewinski

**Affiliations:** 1Department of Chemical and Life Science Engineering, Virginia Commonwealth University, Richmond, VA 23284, USA; tians2@vcu.edu (S.T.); stevensr2@vcu.edu (R.S.); 2Department of Computer Science, Virginia Commonwealth University, Richmond, VA 23284, USA; btmcinnes@vcu.edu

**Keywords:** machine learning, artificial intelligence, classification, regression, random forest, extrusion-based bioprinting, 3D bioprinting, alginate, gelatin, 3D printing

## Abstract

Optimization of extrusion-based bioprinting (EBB) parameters have been systematically conducted through experimentation. However, the process is time- and resource-intensive and not easily translatable to other laboratories. This study approaches EBB parameter optimization through machine learning (ML) models trained using data collected from the published literature. We investigated regression-based and classification-based ML models and their abilities to predict printing outcomes of cell viability and filament diameter for cell-containing alginate and gelatin composite bioinks. In addition, we interrogated if regression-based models can predict suitable extrusion pressure given the desired cell viability when keeping other experimental parameters constant. We also compared models trained across data from general literature to models trained across data from one literature source that utilized alginate and gelatin bioinks. The results indicate that models trained on large amounts of data can impart physical trends on cell viability, filament diameter, and extrusion pressure seen in past literature. Regression models trained on the larger dataset also predict cell viability closer to experimental values for material concentration combinations not seen in training data of the single-paper-based regression models. While the best performing classification models for cell viability can achieve an average prediction accuracy of 70%, the cell viability predictions remained constant despite altering input parameter combinations. Our trained models on bioprinting literature data show the potential usage of applying ML models to bioprinting experimental design.

## 1. Introduction

Three-dimensional (3D) bioprinting is a bottom-up fabrication approach to create tissue-mimetic structures through the precise deposition of biomaterials. Extrusion-based bioprinting (EBB) is a subset of this technique, dispensing biomaterials through a container using pressure exerted pneumatically or mechanically. Due to its ability to produce constructs with the desired structural stability, precision in microstructure creation and cellular arrangement, and versatility in biomaterial, cell density, and additive usage, EBB has emerged as a leading technology for the regenerative medicine field. In particular, EBB has been utilized for damaged organ or tissue replacement or in vitro tissue creation for drug development and disease modelling. 

When live cells are embedded within the biomaterials used, a combination of material parameters and printer settings impact the cells’ viability when extruded, including nozzle outlet diameter, material concentration, and operating temperature. In addition, these parameters affect the ability of the biomaterials to produce precise geometries, also known as printability. Thus far, the optimization of EBB parameters has been mainly conducted through systematic wet-lab experimentation. This process can be laborious, and the information gained can be difficult to translate towards different biomaterials and printers. 

A potential solution to expedite EBB experimental design is through machine learning (ML). ML is a subset of artificial intelligence models that can analytically identify relationships among input parameters and predict desired outcomes based on said relationships. The application of ML to improve additive manufacturing processes has been well documented, including the predecessor technology to EBB, fused deposition modeling (FDM) [1]. ML has mainly been used to optimize printing and geometric design parameters to further improve material properties [2,3,4] and optimize material usage [5]. Structural and geometric detection have also been aided through the use of ML through optical [6] and acoustic sensors [7]. The inclusion of ML in 3D bioprinting is relatively new, although unique contributions have been made thus far. Shi et al. implemented a multilayer perceptron-based artificial neural network trained with computational fluid dynamics simulations of droplet formation and flow behavior to predict classification-based droplet behavior using voltage, nozzle diameter, bioink surface tension, and bioink viscosity input parameters for a drop-on-demand bioprinting system [8]. Experimental validation of six different input parameter combinations that were predicted to produce single, satellite, or no droplets confirmed for each case that experimental results matched with droplet formation predictions. The same group developed a multi-objective optimization design method using gradient descent-optimized, fully connected neural networks to form single droplets based on optimized voltage, nozzle diameter, bioink viscosity, and bioink surface tension in comparison to randomly set voltages, bioinks with arbitrary surface tensions and viscosities, and printer nozzle diameter [9]. Specific to EBB, ML has been used for iterative optimization of printability. Lasso regression was used to optimize printed structures in a support bath structure using both the underlying physical parameters that are not directly manipulated along with the directly manipulated experimental variables [10]. The benefit of using this model was that a specific combination of construct height, support bath material concentration, and retraction distance was found to retain print fidelity while printing at a faster speed. Another iterative study applied Bayesian Optimization on an initial dataset of printability scores based on material and EBB printing parameters, of which parameter combinations were predicted with new experimental results to improve printability scores until an optimal parameter combination was met with the highest possible printability score [11]. This process resulted in needing 4 to 47 experiments to find optimal parameter combinations compared to using a total possible number of experiments ranging from 6000 to 10,000 determined by the Bayesian Optimization algorithm. Conev et al. also examined random forest regressor and classifier capabilities in determining printed construct quality using a previous EBB dataset containing systematic examination of poly(propylene fumarate) [12,13]. Results indicated satisfactory labeling performance from both random forest models. The common theme amongst the above studies is that living cells were not used. Incorporating cellular parameters and predicting cellular performance in bioprinted constructs appears to be the next step in ML incorporation in bioprinting. Lee et al. tested cell viability based on collagen, hyaluronic acid, and fibrin formulations predicted using the relative least general generalization algorithm along with multiple regression modeling for printability [14]. On top of maintaining suitable shape fidelity, cell-laden scaffolds with optimized material concentrations exhibited increasing cell proliferation and migration up to 28 days after printing. Xu et al. developed a model based on ensemble learning for cell viability prediction in stereolithography-based bioprinting [15]. Prediction performance on 10% of the dataset used showed a coefficient of determination (R^2^) score of 0.953, indicating high goodness-of-fit for viability prediction of new parameter combinations. 

In this study, we applied ML to assist in experimental designs of alginate/gelatin-based hydrogel with cells, of which we will refer to as bioink. The database we utilized contains bioink material concentration, solvent used, polymer crosslinking information, printing settings, cell viability, and printability results accrued from 75 EBB manuscripts over the past 13 years. Shown through previously mentioned studies, data used for bioprinting ML model training and testing has only been gathered from and applied within group. To our knowledge, our compilation of experimental data and parameters reported from different bioprinting laboratories for ML applications is the first of its kind. The database contains 617 unique instances of cell viability and 339 unique instances of printability. We analyzed the ability of ML regression and classification techniques to accurately and precisely predict cell viability and printability outcomes based on certain combinations of material, biological, and printer parameters. Furthermore, we used the trained models to provide cell viability and printability predictions based on input parameters for an extrusion based bioprinter. 

## 2. Materials and Methods

### 2.1. Dataset Creation

A datasets of 617 instances corresponding to a unique cell viability value and a dataset with 339 instances corresponding to a unique filament diameter value were collected from 75 EBB papers found through the search terms *TS = Extrusion AND (Bioprinting OR Bioink)* and *TS = (Extrusion OR Extrud*) AND (Bioprint* OR Bioink*) AND (alginate*) AND (gelatin*) AND (viability OR viable* OR surviv* OR death OR proliferat*)* in Web of Science. Material concentration, solvent usage, crosslinking mechanism and duration, printer settings, observation duration, cell viability, and filament diameter were recorded for each unique instance of either cell viability and/or filament diameter. When cell viability data was presented in graphical form, PlotDigitzer software (http://plotdigitizer.sourceforge.net, last accessed on 22 May 2021) was used to estimate cell viability values in relation to the viability scales they were presented against. Filament diameter values were extracted (via PlotDigitizer) from images provided in different manuscripts corresponding to different times of observation after printing. The datasets created are available through the Open Science Framework [16]. 

### 2.2. Machine Learning Experimental Design 

We framed the prediction of cell viability, filament diameter, and extrusion pressure as supervised regression-based and classification-based questions. In the regression models, a value of cell viability and filament diameter was predicted based on the training set and compared with the true cell viability and filament diameter values of the test set. For cell viability classification models, a binary class was created from the numerical cell viability data by setting a threshold for acceptable cell viability to be equal to or above 80.0%. The cell viability class was “Acceptable Cell Viability” with values of “Y” for yes and “N” for no. For filament diameter classification models, a binary class was created from the numeric filament diameter data by setting a threshold for tolerable filament diameter equal or above 10.0% error [17,18]. This was determined by calculating the absolute difference between filament diameter and nozzle diameter and dividing by nozzle diameter. The class was named “Acceptable Filament Diameter” with values of “Y” for yes and “N” for no based on above criteria. At hydrostatic pressures above 100 kPa, cell metabolic behavior can become negatively affected [19]. In cell viability instances with stated extrusion pressures, instances with a pressure above 100 kPa were deemed to have unacceptable extrusion pressure, while the rest were deemed acceptable. We evaluated three regression learners in this study: (1) support vector regression, (2) linear regression, and (3) random forest regression; and three classification learners: (1) random forest classification, (2) logistic regression classification, and (3) support vector machines. 

#### Evaluation

Metrics used for evaluating regression model performance were the coefficient of determination (R^2^) and mean squared error (MSE). R^2^ is a measure of goodness of fit of the model on provided data. It indicates the proportion of the variance in the dependent variable that is explained by independent variables. A perfectly fit model will have a R^2^ value of one. MSE indicates the average of the squares of errors. Errors are the differences between actual values and predicted values. As MSE values become closer to zero, the lower the overall error becomes for model fit onto data. One regression model was chosen for prediction usage based on the highest coefficient R^2^ values and lowest MSE over k-fold cross validation training evaluation up to k = 10 relative to other models. 

Metrics used for evaluating classification model performance were accuracy, precision, and recall scores. Accuracy represents the percentage of correctly predicted outcomes for a sample. Precision is calculated by the ratio of true positive prediction over the sum of true positive and false positive predictions. Precision represents the proportion of correct predictions over sum of all predictions of the same label. Recall is similar to precision calculations, but the amount of false positives is replaced by false negatives in predictions. This value represents the proportion of all instances of the same label that are predicted correctly. Overall, a classification model was chosen for prediction usage based on the highest average prediction accuracy over k-fold cross validation training evaluation up to k = 10. 

Chosen models were then utilized to predict acceptable cell viability and filament diameter from material and printing parameter combinations feasible to conduct in our laboratory for experimental verification of the predicted values. In addition, extrudability of low viscosity and high viscosity bioink was also tested using materials and material concentrations within range of the dataset by predicting the extrusion pressure that would produce the desired cell viability and filament diameters. 

### 2.3. Data Preprocessing 

Within the dataset, null instances for bioink temperature (i.e., syringe temperature) and printing substrate temperature were set at 22 °C as the majority of experiments were conducted or were assumed to be conducted at room temperature. Additional variables with more than 50% null values were removed from the dataset and non-printing instances were also removed (instances with cast molded bioink or other methods with cells cultured within non-extruded hydrogel). Variables with only null instances and instances of zero units were removed prior to model usage as available imputation methods of null values would not provide an accurate representation of actual quantitative values of the variables used in respective manuscripts. Additional variables with null values and non-zero instances were imputed through k-nearest-neighbors imputing with a neighbor range of 30. Categorical data was encoded through one-hot-encoding. Feature selection was performed by conducting feature importance analysis on variables within the cell viability and filament diameter datasets using random forest regression. For regression model performance evaluation, continuous variable instances were normalized through the MinMaxScaler function (Sci-kit Learn package, Python 3.7).

### 2.4. Dataset Training Size Variation

Cross-validation of datasets was used to test training size variation by varying how many folds the training data was divided into. The greater the number of folds, the greater the number of instances used for training. For each model, performance metrics were compared by k-fold cross validation with *k* values of 2, 5, and 10. 

### 2.5. Intrastudy Model Creation and Usage

A comparison of general dataset predictive ability was done with a selected study that used alginate and gelatin multicomponent hydrogels [20]. Sixteen instances of unique cell viability outcomes from material and equipment parameters were used to create a random forest classification and regression model, as well as a linear regression and support vector regression model for cell viability. Filament diameter trend was also produced from four filament diameter data points corresponding to different material and pressure combinations through multiple regression. Two cell viability values, one based on parameter values within range of the intrastudy dataset, and another based on parameter values out of range of the intrastudy dataset, were predicted for and compared against predicted values of the overall dataset. Filament diameter of constructs printed with an alginate and gelatin multicomponent bioink was compared against the intrastudy regression model as well as with the random forest regression model predictions made for the same material and equipment parameters. Since only 4 filament diameter values were provided in the specific study, a fitted regression model was used. A multiple linear regression was fit to data correlating extrusion pressure and alginate concentration with filament diameter, resulting in a regression equation, Equation (1), of:*z* = *Ax* + *By* + *C*(1)
where *A* = 333.26, *B* = −0.245, and *C* = −781.4. The variable *x* represents the alginate concentration (% *w*/*v*), *y* is the extrusion pressure (kPa), and *z* is the filament diameter (μm). 

### 2.6. Material Preparation

Sodium alginate powder (Sigma W201502, St. Louis, MO, USA) and gelatin (type B, 300 bloom derived from bovine, Sigma G9382, MO, USA) were sterilized under UV radiation for 30 min. Afterwards, the powders were dissolved in complete cell culture media composed of Dulbecco’s Modified Eagle Medium (DMEM, Gibco, Grand Island, NY, USA), 10% fetal bovine serum (FBS, Life Technologies, Grand Island, NY, USA) and 1% penicillin-streptomycin (Gibco, Grand Island, NY, USA). The mixtures were heated to 50 °C and magnetically stirred for 4 h. Complete mixtures were then vortexed for 1 min and centrifuged at 3000 rpm for 3 min to eliminate bubbles. Hydrogels were stored at 4 °C prior to experimentation. Concentrations of sodium alginate (Alg) and gelatin (Gel) mixtures in complete media are denoted as Alg/Gel in units of % *w*/*v*. The extrusion of bioinks and biomaterial inks were conducted at 22.5 °C. The 100 mM CaCl_2_ solution used to crosslink printed constructs was prepared by dissolving CaCl_2_ (Sigma-Aldrich, St. Louis, MO, USA) in complete cell culture media and sterile filtering through a 0.22 µm syringe filter (Millipore, Cork, Ireland).

### 2.7. Cell Culture

Mouse neuroblastoma cells (N2A, CCL-131 cell line, American Type Culture Collection, ATCC) were cultured at 37 °C in humidified 5% CO_2_ atmosphere using complete cell culture media in T75 cell flasks (Falcon™, Corning, Durham, NC, USA). Cells were passaged every 4 to 5 days with 0.05% trypsin/EDTA (Gibco, Grand Island, NY, USA), and a portion was split for use to prepare bioinks for printing.

### 2.8. Construct Bioprinting

Alg/Gel hydrogels were heated up to 37 °C prior to mixing with cells. Cell suspensions containing 1.0 × 10^6^ trypsinized cells were centrifuged to create cell pellets for mixing. A cell density of 1.0 × 10^6^ cells/mL was chosen due to it being the most common cell density reported amongst studies used to compile the training dataset. To this cell pellet, 1 mL of liquified hydrogel was added using a 10 mL syringe (BD Falcon, Franklin Lakes, NJ, USA) and then triturated using a pipet for 30 s to mix thoroughly. The mixture was then aspirated into a 10 mL syringe and transferred to a 3 mL cartridge (Nordson EFD, East Providence, RI, USA) via a female-to-female luer lock connection. The bioink was then held at room temperature to allow for complete gelation. 3/4 Alg/Gel was held for 90 min, 3/7 Alg/Gel was held for 30 min, and 8/20 Alg/Gel was held for 20 min at room temperature after mixing with cells at 37 °C. The duration of complete gelation depends on the concentration of sodium alginate and gelatin used. Once gelation was reached, the 3 mL cartridge was then secured onto an extrusion-based bioprinter (INKREDIBLE, Cellink, Boston, MA, USA). For cell viability testing, 80 mm × 80 mm × 0.8 mm models were printed with 22G conical nozzles (Nordson EFD, East Providence, RI, USA) at a feed rate of 10 mm/s into 24 well cell culture plates at 22.5 °C. For confocal microscopy imaging, models were printed onto sterile cover glass slides. Directly after printing completion, pictures of constructs were taken, and constructs were exposed to 100 mM CaCl_2_ crosslinking solution for 1 min. Afterwards, remaining crosslinking solution was aspirated and constructs were rinsed with Dulbecco’s PBS (DPBS, 7.4 pH, Gibco, Paisley, UK). The constructs were then incubated at 37 °C at 5% CO_2_ with complete cell culture medium. 

### 2.9. Live/Dead Staining

N2A cell viability was determined by staining cells with Hoechst 33342 (40.6 µM) and propidium iodide (19.7 µM) dye solutions (Readyprobes, ThermoFisher, Eugene, OR, USA) following the manufacturer’s protocol. Briefly, cell culture media was aspirated and replaced with DPBS containing 1 drop of Hoechst 33342 and 1 drop of propidium iodide and then incubated for 15 min at 37 °C at 5% CO_2_ with no light exposure. Excitation/emission wavelengths of 358/461 nm and excitation/emission wavelengths of 580/604 nm were used to image Hoechst 33342 and propidium-iodide-stained cells, respectively, using an imaging plate reader (Cytation 3, BioTek, Winooski, VT, USA). Z-stack images of stained cells in bioink were taken through confocal microscopy (LSM 710, Zeiss, Jena, Germany). Magnification of plate reader images was set at 4× while confocal microscopy image magnification was set at 10×. Cell counting for cell viability was conducted using Cytation 3 Cell Imaging software. Cell viability was determined by dividing the total number of cells (total number of Hoechst 33342-stained cells subtracted by the number of dead cells stained from propidium iodide) by the total number of Hoechst 33342-stained cells. 

### 2.10. Filament Diameter Measurements

Constructs were imaged using an imaging plate reader (Cytation 3, BioTek, Winooski, VT, USA). Collected images were analyzed using ImageJ (https://imagej.nih.gov/ij/, last accessed on 22 May 2021) for filament diameter length.

### 2.11. Extrusion Pressure Measurements

Using cell viability dataset instances with available extrusion pressure values (353 instances)*,* random forest regression and linear regression models were created to predict extrusion pressure values needed to extrude specific material concentrations to produce 80% cell viability. Bioinks of 3/4 Alg/Gel, 3/7 Alg/Gel and 8/20 Alg/Gel were used to test extrusion pressure predictability within and near the edge of material concentration bounds of the dataset used. Material preparation procedure for testing extrusion pressure was the same as in Section 2.8.

### 2.12. Statistical Analysis

Cell viability, filament diameter, and extrusion pressure measurements were expressed as mean ± standard deviation. Statistical significance between any two groups of either cell viability, filament diameter, and extrusion pressure measurements were tested through one-way ANOVA, with the significance level set as *p* < 0.05. Percent error was calculated for experimental cell viability, filament diameter, and extrusion pressures as compared to predicted values.

## 3. Results

### 3.1. Performance of Different Regression and Classification Models

#### 3.1.1. Model Performance on Cell Viability

Amongst regression models, the random forest regression models for cell viability predictions elicited higher R^2^ values while minimizing the average MSE (Figure 1). 

The random forest classification models elicited higher prediction accuracy, precision, and recall than other models tested (Figure 2). 

Both the logistic regression and support vector classification models elicited the same performance values for accuracy, precision, and recall due to labeling all cell viability classifications as acceptable cell viability during the model fitting process (Appendix A). Feature importance testing based on decision trees generated from the random forest tree models indicated relatively major effects from extrusion pressure, specific material concentration, solvent choice, nozzle diameter, and printing temperatures for cell viability predictions (Figure 3).

#### 3.1.2. Model Performance on Filament Diameter

Amongst regression models, random forest regression models for filament diameter predictions also produced higher coefficients of determination while minimizing average mean squared error (Figure 4). 

Similar to cell viability classification, when predicting acceptable filament diameter, the random forest classification models produced higher prediction accuracy, precision, and recall than the other models tested (Figure 5). 

The support vector classification model generated precision and recall scores of zero due to labeling all filament diameter tolerance classifications as out of tolerance during the model fitting process (Appendix A). Through feature importance analysis, nozzle diameter was ranked as the most important feature affecting filament diameter model prediction (Figure 6).

#### 3.1.3. Model Predictions Compared to Experimental Trends (Non-Primary Cells)

Holding all but one input parameter constant, including the time of observation at zero days, both the random forest and linear regression models translated the impacts of several physical variables onto prediction trends. The regression models predicted decreased cell viability with increasing alginate concentration, increasing syringe temperature above 37 °C, or increasing extrusion pressure. The linear regression model also discerned trends reflective of the dataset. Specifically, lower extrusion pressures resulted (1) in higher cell viabilities, (2) when increasing syringe or cartridge temperature, (3) when increasing cell density, (4) with lower gelatin concentrations, and (5) with larger nozzle diameters.

When predicting filament diameter, the random forest regression model predicted decreasing filament diameters (1) when ionic crosslinking duration post-extrusion is above 9 min, (2) when extrusion pressure is increased up to 90 kPa, and (3) when nozzle diameter is decreased. 

The linear regression model further predicts smaller filament diameters when syringe temperature, printing substrate temperature, gelatin concentration, CaCl_2_ concentration, and ionic crosslinking duration increased individually. Furthermore, the filament diameter increased with increasing alginate concentration when predicted with linear regression. Using the random forest classification, the filament diameters produced were deemed to be within tolerance when using nozzle diameters of 840 μm or larger.

#### 3.1.4. Effect of Training Data Size on Cell Viability and Filament Diameter Predictions

Through increasing the number of cross validation folds, R^2^ increased while MSE performance saw minimal change for two random forest regression and linear regression on cell viability predictions (Appendix A). For random forest cell viability classification, we can see that accuracy, precision, and recall stayed consistent with increased number of folds and in turn, training set size for cell viability (Appendix A). For filament diameter modeling, random forest regression model saw minimal effects due to training data size, while linear regression saw large increases in R^2^ and decreases in MSE as the number of cross validation folds increased from two to five (Appendix A). Accuracy, precision, and recall did not see significant changes regardless of increasing training data size (Appendix A).

### 3.2. Effect of Specified Training Data on Cell Viability and Filament Diameter 

#### 3.2.1. Effect on Cell Viability Predictions

Using the complete cell viability dataset for model training, the random forest regression model resulted in a predicted cell viability of 73.1% for a material combination of 3/4 Alg/Gel, 100 mM CaCl_2_ crosslinking solution with an exposure duration of 60 s, and extrusion through a 22G conical nozzle at room temperature (22.5 °C). For another material combination of 3/7 Alg/Gel, 100 mM CaCl_2_ crosslinking solution with an exposure duration of 60 s, and extrusion through a 22G conical nozzle at room temperature (22.5 °C), the random forest regression model predicted the same cell viability value of 71.7%. 

A specific study was used to create an alginate- and gelatin-focused dataset for random forest regression model training [17]. For 3/4 and 3/7 Alg/Gel, a random forest regression model created from this specified dataset resulted in a cell viability prediction of 91% for both material combinations when the extrusion pressure was set constant. The actual cell viability of values gathered from live/dead staining showed a larger amount of dead cells present directly after printing in 3/7 Alg/Gel than in 3/4 Alg/Gel constructs (Figure 7 and Appendix A). 

The resultant cell viability values for the 3/4 and 3/7 Alg/Gel constructs are 85.2 ± 9.1% and 64.2 ± 10.6%, respectively (Table 1). The random forest classification, logistic regression, and support vector regression models predicted acceptable cell viability for both material conditions based on tested material concentration and printing parameters (Appendix A). All predictions were made keeping extrusion pressure constant at 95.4 kPa.

#### 3.2.2. Effect on Filament Predictions 

Using the complete filament diameter dataset for model training, the random forest regression model resulted in predicted filament diameters of 1073 μm and 857 μm for 3/4 and 3/7 Alg/Gel, respectively (Table 2). Filaments from the constructs printed with the 3/4 Alg/Gel resulted in 1157 ± 102.2 μm diameter pre-crosslinking and 927.6 ± 106.0 μm diameter after crosslinking. For the 3/7 Alg/Gel, the filament diameter pre-crosslinking was measured at 817.0 ± 107.7 μm, while measuring at 707.2 ± 146.1 μm directly after crosslinking (Figure 8 and Figure 9). 

The percent errors of the crosslinked filament diameters with respect to nozzle diameter (410 μm) is 126% and 72.5% for 3/4 Alg/Gel and 3/7 Alg/Gel constructs, respectively, making the filament diameters out of tolerance. All classification models predicted unacceptable filament diameter tolerance for both the 3/4 Alg/Gel and 3/7 Alg/Gel combinations (Appendix A).

### 3.3. Extrusion Pressure Recommendation Predictions 

Both random forest regression and linear regression models indicated increased pressure needed with higher alginate and gelatin concentrations, although the random forest regression model predicted a lower range of extrusion pressures, while the linear regression model predicted a higher range. Based on feature importance rankings, substrate temperature appears to be the most significant variable impacting extrusion pressure used (Figure 10). 

Constructs printed using 3/7 Alg/Gel required an average extrusion pressure of 71.7 kPa. Bioink with 8/20 Alg/Gel was not able to form printed constructs due to high material viscosity, although over-deposited filament was extruded at an average pressure of 208.3 kPa. As material concentrations increased, the prediction accuracy of the random forest regression diminished while the prediction accuracy of the linear regression improved, as noted by percent error calculations (Table 3). The random forest classification model was able to predict acceptable extrusion pressure correctly for the 3/4 Alg/Gel and the 8/20 Alg/Gel, but not for the 3/7 Alg/Gel. Meanwhile, logistic regression and support vector classification models predicted that all material concentration combinations printed under the same printing settings can result in using pressure within the acceptable pressure range (Appendix A). All model predictions were conducted with desired cell viability set to 90% immediately after printing. In the cases of 3/4 Alg/Gel and 3/7 Alg/Gel, the pressure needed for the extrusion and construct formation was smaller than predicted and the resulting cell viabilities were also lower than 90% (Table 1).

## 4. Discussion

In this study, we approached the application of ML to bioprinting in two ways. First, we applied regression and classification models to data derived from a single study, which is more directly comparable to published studies on ML in bioprinting. In addition, we went further and applied the same ML techniques to a larger dataset encompassing results from 75 different studies to understand whether this data aggregation approach can effectively widen the area of model applicability. The random forest regression, random forest classification, and linear regression models created can be used to an extent in conjunction with one another for outcome prediction as well as condition recommendation. Specifically, the random forest classification models for cell viability and filament diameter predictions were able to generate similar average model accuracy scores compared to previous literature’s model performance accuracy also using random forest models [12]. Both the random forest and linear regression models, more so the linear regression models, have shown the ability to represent several physical phenomena that have been documented in previous bioprinting or hydrogel studies. In particular, the general trends of increased extrusion pressure and alginate concentration resulting in decreased cell viability prediction values correlates with findings in previous literature that indicate increasing alginate concentration results in decreased cell viability [21,22,23]. In other cases, trends of predicted values oppose what is seen in the literature [24,25]. These trends include: (1) decreased cell viability with increasing nozzle diameter, (2) increased cell viability with increasing gelatin concentration, and (3) larger filament diameter in DMEM-based bioink compared to saline solution-based bioink.

Feature importance ranking results indicated that cell density as a parameter did not carry as great of a weight in the random forest predictive function for cell viability compared to the other bioink and equipment parameters. Increasing cell density in bioink has been shown to marginally improve cell viability in the short term (0 to 1 day post-printing) for primary cells and stem cells [26,27]. Increasing cell density may also lead to an increase in cell agglomerates. In cases with cell densities above 5.0 × 10^6^ cells/mL, cell viability can decrease drastically the longer printed constructs are cultured (7 to 21 days) [27]. This can be due to the creation of hypoxic conditions for cells in inner areas of cell agglomerates, which limits nutrient and waste transport through cell structure. In the cell viability dataset, the correlation of cell density with cell viability does not result in notable trends when cell density increases. Amongst the 617 instances used for cell viability model training, only 196 instances used cell densities above 5.0 × 10^6^ cells/mL. Within those instances, 65.3% of cell viabilities are acceptable (≥80%). This is a similar distribution to the cell viability value distribution in the overall dataset, where 61.6% of cell viability values are deemed acceptable (≥80%). In addition, the majority of unacceptable (<80%) cell viability amongst instances containing more than 5.0 × 10^6^ cells/mL corresponded with cell density values between 5.0 to 10.0 × 10^6^ cells/mL, while instances with higher cell concentrations saw smaller portions of cell viability values being unacceptable.

Compared to other ML models created for bioprinting predictions, the regression models created in this study provided lower R^2^ values and comparable errors with a similar proportion of training data to test data, and the accuracy of the classification models were lower as well [12,15]. A major reason for this is the difference in experiment variation for the datasets used to create the models. Input parameters gathered from published studies contained a limited number of independent variables due to the chosen experimental design, which focused on answering a specific research question versus parameter optimization. In addition, our dataset is inherently heterogeneous due to being acquired from studies conducted using different testing conditions and printing strategies. Comparatively, past studies contained larger amounts of data collected from controlled experimental settings [12]. Not all of the input conditions used in developing the ML models were reported in every study included in the dataset. Although missing data can be estimated using imputation, this can lead to a misrepresentation of the features’ weight on the output parameters and consequently lead to worse performance metrics along with prediction values that do not correlate with experimental results. 

Previous EBB studies have shown that increasing pressure-induced shear stress on cells can cause decreases in cell viability for both immortalized cell lines and stem cells [17,28,29]. In a case with 10% *w*/*v* gelatin printed with HepG2 cells and 27 gauge conical nozzles, cell viability notably decreased from 96% to 84% when the pressure increased from 200 to 300 kPa [28]. Blaeser et al. indicated a notable decrease in the cell viability of L929 fibroblasts encapsulated in alginate hydrogels when the average shear stress within the printing orifice reached 5 kPa or above [29]. Specifically, cell viability dropped from 96% in cases with less than 5 kPa average shear stress to 91% cell viability within 5 to 10 kPa and dropped further to 76% at higher shear stress values. In terms of pressure, a 5 kPa shear stress value corresponded to a pressure between 100 and 150 kPa when a 300 µm cylindrical valve is used along with an alginate concentration of 1.0% *w*/*v*. Since N2A cells were used in the validation experiments, the cell viability behavior under shear stress would be similar to previous studies also using non-primary cell lines. The random forest classification was seen to produce varied prediction results in cases of primary cell usage with conical nozzles. Testing the predicted effects of extrusion pressure on primary cells printed through conical nozzles, cell viability was found to become unacceptable above 20 kPa for 3/5 Alg/Gel, while 3/8 Alg/Gel was found to have acceptable viability across pressures from 0 to 300 kPa. Increasing alginate concentrations, 5/2 Alg/Gel also saw unacceptable cell viability above 20 kPa, while 5/4 Alg/Gel saw unacceptable viability only when above 270 kPa. When varying the syringe cartridge temperature for printing primary cells, the 3/5 Alg/Gel bioink saw unacceptable cell viability at temperatures above 20 °C, while 3/8 Alg/Gel usage resulted in unacceptable cell viability at 36 °C or above. Interestingly, a 5/2 Alg/Gel material concentration resulted in unacceptable cell viability specifically at 23 °C as well as at temperatures above 36 °C, while all other temperatures from 4 to 40 °C resulted in an acceptable cell viability. When gelatin concentration increased to 4% *w*/*v* while alginate concentration remained constant at 5% *w*/*v* (5/4 Alg/Gel), all predicted cell viability values up to 40 °C were acceptable. In the case of predicting suitable extrusion pressure, the use of primary cells resulted in a decrease of around 20 kPa less pressure needed for the same material concentration and printing setting as compared to using non-primary cells. Overall, to elucidate more straightforward modeling of primary cell viability behavior, more data gathered from studies using primary cells and straight nozzles is needed to understand if nozzle geometry imparts different biological effects for primary cells compared to non-primary cells. 

Overall, the random forest regression models for cell viability, filament diameter, and extrusion pressure resulted in predictions based on grouping. For example, when setting all input parameters to constant except for gelatin concentration, the extrusion pressure was predicted to be 233.9 kPa for parameter combinations where gelatin concentration is at 5% *w*/*v* or above. This phenomenon is due to the random forest regression model’s tree-based training approach, where numerous decision trees with different decision paths are created and ensembled together to extract average prediction values from all trees’ prediction outcomes. The function of the random forest regression models also contributes to the much higher R^2^ value compared to the linear and support vector regression models trained on the models as well. Since the decision trees created through random forest regression are limited in the number of subsequent decision nodes, the predicted values across different trees, when ensembled, can have values in very small ranges for different decision nodes, which are then grouped together as the type of outcome. In this way, the model can result in higher goodness-of-fit values than other curve-fit regressions. Comparatively, using a curve fit-based regression model such as a linear regression or a support vector regression did not output the same grouping phenomenon, but showed continuous data change. Linear regression models also provided very comparable mean squared errors to random forest regression.

Models built from data collected from one study did not seem to be enough to generate accurate predictions. The specific study used resulted in cell viability and filament diameter predictions that produced greater percent errors relative to the experimental values as compared to the models built on entire datasets. This is due to the study containing only certain ranges and values of input parameters. When using models to predict outcomes with conditions out of range of the training data, defaulting to the most common prediction outcome or extrapolation will occur in tree-based or curve-fit based models, respectively (Table 1). Unless the range of input parameters used for predictions lies within the range of the trained model, data from multiple studies that cover the required ranges are needed.

For cell viability predictions, regression models hold promise for further development. The random forest filament diameter regression model offers greater prediction accuracy compared to the linear regression model based on the percent error from actual filament diameter values. If the user knows the extrusion range suitable for their bioink, the filament diameter predictions can become even more accurate (Table 2). Amongst all models, the filament diameter prediction models mapped the closest to experimental results when accounting for the prior knowledge of suitable pressure ranges to input for predictions. Unlike the other predictive models, the nozzle diameter and extrusion pressure were relatively much more impactful variables to the model (Figure 6) as compared to the most important variables found through feature importance of the other random forest models (Figure 3 and Figure 10). For extrusion pressure predictions, the random forest regression model underestimates the required extrusion pressure, while the linear regression model overestimates the required pressure. The correction factors determined from the uncertainty factor evaluation can be applied for these models to produce prediction outcomes closer to the actual results. For the models in this study to be used effectively, users still need to have a baseline knowledge of how material parameters and printing settings affect cell viability, filament diameter, and extrusion pressure needed, such as in the case of the filament diameter regression models. Based on the trends extracted from tuning different parameters, future experiments could focus on collecting more data for the variables to improve the predictive power of the models. 

The nature of how cell viability values are derived and calculated can play a large role in how representative they are of true biological conditions of cells within printed constructs. In studies using live/dead staining to derive cell viability values, how large the area of focus is on the construct for cell counting is accounted for. A standard area of observation for a section of the construct filament is not provided. In most cases, it is not clear at what focus the transverse plane of a filament is examined. Furthermore, whether specific sections of a construct are used (e.g., the outer-boundary filament strands at an intersection of filament in the middle of a construct) or randomly selected sections are selected for cell viability measurements is not clear. 

Cell viability can be determined by measuring the different endpoints, such as cell membrane integrity, metabolic activity, and mode of death (apoptosis versus necrosis). A majority of EBB studies utilize dye exclusion live/dead staining [30]; therefore, the cell viability values in the dataset created for this study can be compared similarly amongst each other. However, grouping cell viability values derived from different assays for model creation may introduce variability due to the measurements of different biological endpoints. Assays that use different mechanisms than live/dead staining dyes used in this study (Calcein AM, propidium iodide, and ethidium homodimer), such as the MTT (3-(4,5-Dimethylthiazol-2-yl)-2,5-Diphenyltetrazolium Bromide) absorbance assay, can provide different relative viability values from colorimetric readings as compared to stained cell counting. Despite variations in the dye combinations and the disparate measurement procedures for live/dead cell staining assays, the random forest regression model’s predicted cell viability values fall within normal experimental ranges. Building upon this study, a future direction can be to compare ML model robustness when trained on data composed of other assays that measure the same cellular endpoints. 

Additional future directions of this work can be to apply experimental results to improve quantitative predictions. The use of first principle calculations can be used to estimate missing variables in the dataset. For example, the Power law or Herschel–Bulkley fluid behavior modeling can be used to find non-Newtonian index values of selected materials to then convert lengthwise and volumetric extrusion rates to missing extrusion pressures, and vice versa [29,31,32,33]. Additional non-linear learners, such as k-nearest neighbor classification and regression models, can be explored as models that generate higher prediction performance than the existing models created. 

## 5. Conclusions

In this study, machine-assisted EBB experimentation was examined through the creation and evaluation of regression-based and classification machine learning models on cell viability, filament diameter, and extrusion pressure. The training data was sourced from literature in the EBB field to understand if different laboratory testing conditions can be synergized for predictive usage under different testing conditions. Results indicated that the generated classification models can elicit suitable accuracy and precision when evaluated on testing data synthesized from literature, while classification and regression models capture physical implications of material and printing settings on outcomes well. Data gathered with a focus on parameters that elicit behavior trends in cell viability, filament diameter, and extrusion pressure can strengthen the database used to produce models that can provide higher accuracy predictions.

## Figures and Tables

**Figure 1 micromachines-12-00780-f001:**
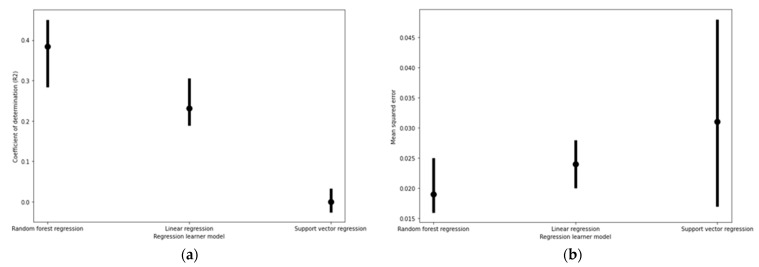
Cell viability regression model performance based on (**a**) coefficients of determination (R^2^) and (**b**) mean squared error (MSE) values under fivefold cross validation. The upper and lower bounds of the error plots represent the maximum and minimum R^2^ and mean square error values produced amongst the five testing and training combinations.

**Figure 2 micromachines-12-00780-f002:**
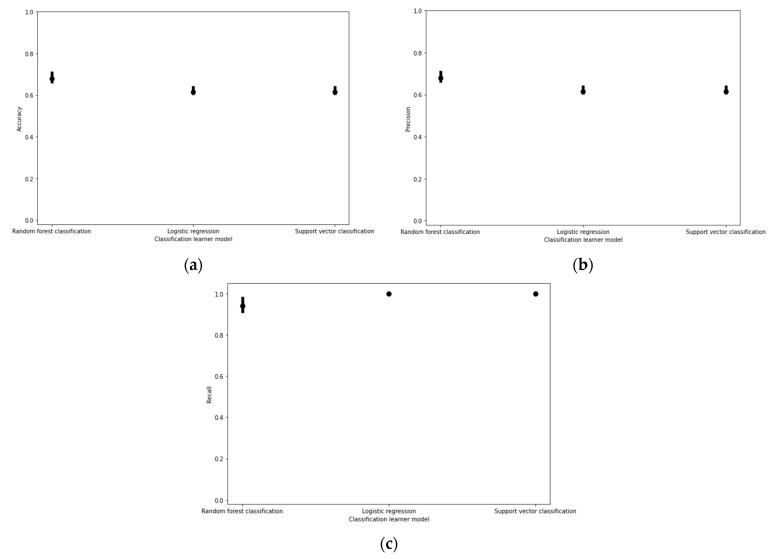
(**a**) Cell viability classification model performance based on accuracy, (**b**) precision, and (**c**) recall scores under fivefold cross validation. The upper and lower bounds of the error plots represent the maximum and minimum scores produced amongst all five combinations of one fold being trained and tested on the remaining four folds.

**Figure 3 micromachines-12-00780-f003:**
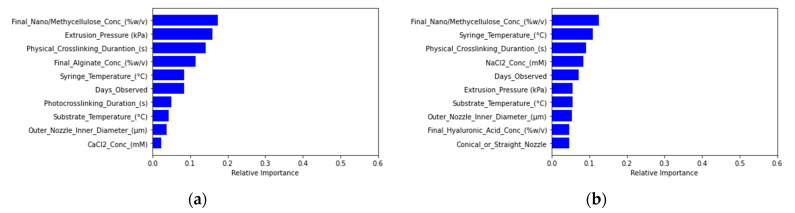
Feature importance rankings of the material, equipment, and experimental parameters based on (**a**) random forest regression and (**b**) random forest classification modeling of cell viability.

**Figure 4 micromachines-12-00780-f004:**
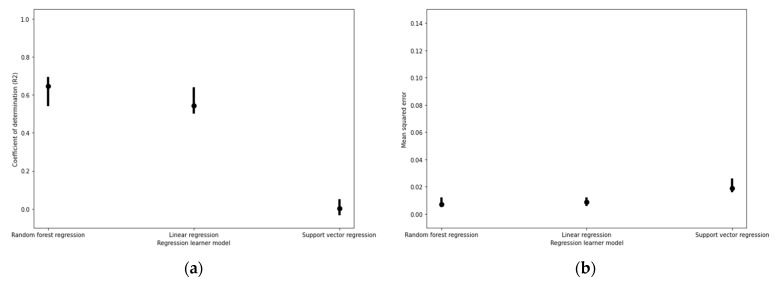
Filament diameter regression model performance based on (**a**) coefficients of determination (R^2^) and (**b**) mean squared error (MSE) values under fivefold cross validation. The upper and lower bounds of the error plots represent the maximum and minimum R^2^ and MSE values produced amongst all five combinations of one fold being trained and tested on the remaining four folds.

**Figure 5 micromachines-12-00780-f005:**
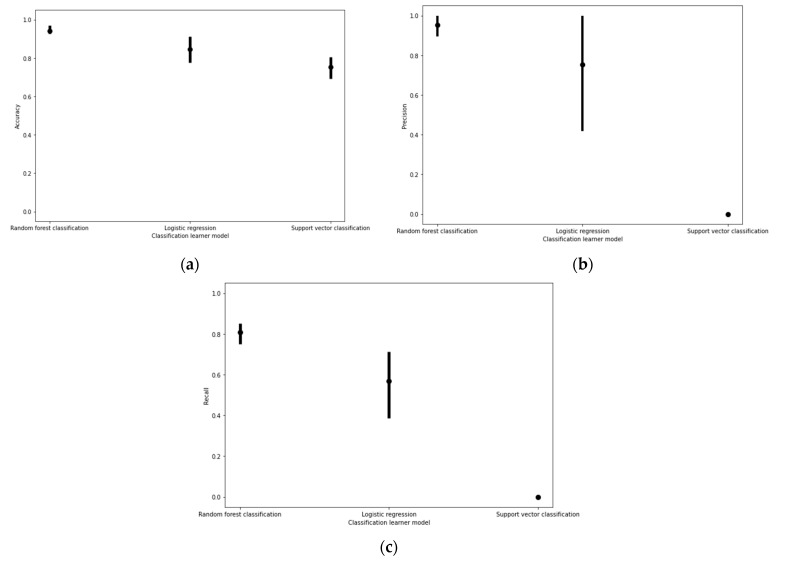
Filament diameter classification model performance based on (**a**) accuracy, (**b**) precision, and (**c**) recall scores under fivefold cross validation. The upper and lower bounds of the error plots represent the maximum and minimum scores produced amongst all five combinations of one fold being trained and tested on the remaining four folds.

**Figure 6 micromachines-12-00780-f006:**
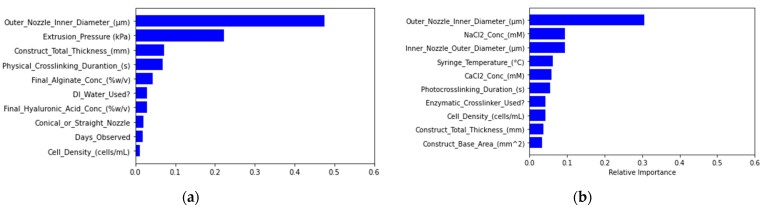
Feature importance rankings of material, equipment, and experimental parameters based on (**a**) random forest regression and (**b**) random forest classification modeling of filament diameter.

**Figure 7 micromachines-12-00780-f007:**
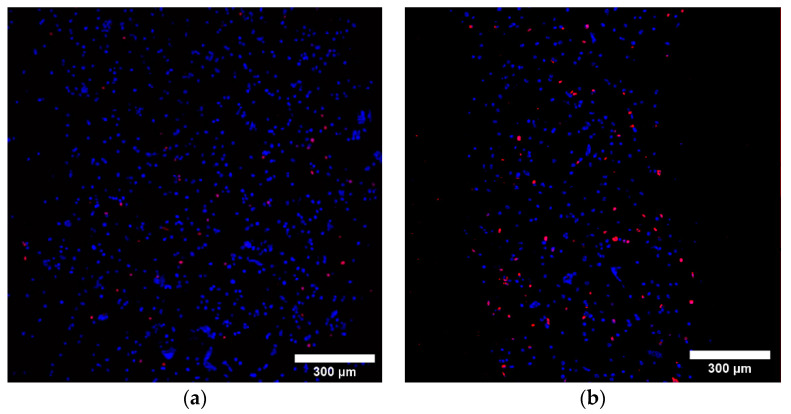
Live/dead fluorescence images (10× magnification) of (**a**) 3/4 Alg/Gel and (**b**) 3/7 Alg/Gel bioinks measured immediately after extrusion. Blue color represents all cells, alive and dead, stained with Hoechst 33342, and red color represents propidium-iodide-stained dead cells. The scale in both figures is 300 µm.

**Figure 8 micromachines-12-00780-f008:**
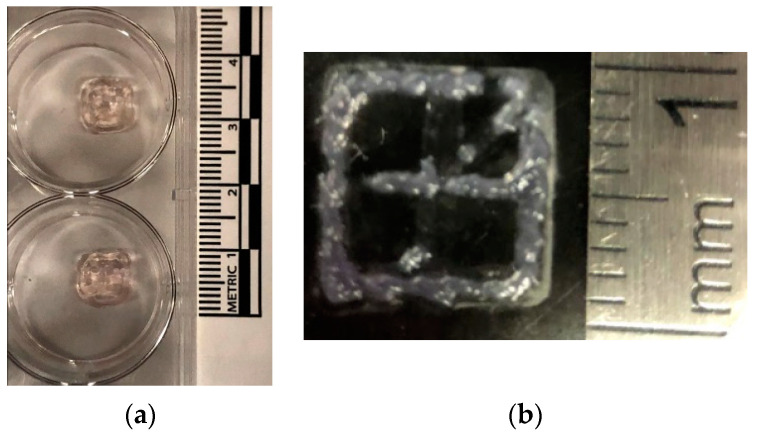
Constructs directly after printing onto tissue-culture treated well plate surfaces (**a**) 3/4 Alg/Gel and (**b**) 3/7 Alg/Gel. The scale in both pictures is 1 cm per unit length.

**Figure 9 micromachines-12-00780-f009:**
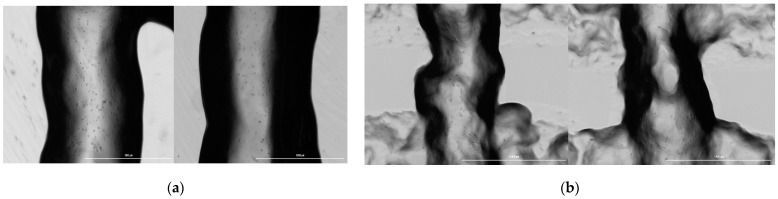
Brightfield images at 4× magnification of (**a**) 3/4 Alg/Gel and (**b**) 3/7 Alg/Gel filaments directly after extrusion. The scale bars depict 1000 µm.

**Figure 10 micromachines-12-00780-f010:**
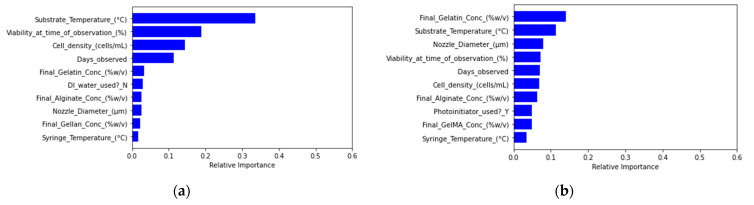
Feature importance rankings of material, equipment, and experimental parameters based on (**a**) random forest regression modeling of extrusion pressure and (**b**) random forest classification modeling.

**Table 1 micromachines-12-00780-t001:** Predicted cell viability values are compared against experimental values for corresponding material concentrations of alginate and gelatin. Actual values represent the mean ± standard deviation for all samples (*n* = number of samples) measured from at least three batches of Alg/Gel bioink.

Prediction Model	Material and Material Concentration (%*w*/*v*)	Predicted Cell Viability (%)	Actual Cell Viability (%)	Error (%)
Random forest regression, complete dataset	3/4 Alg/Gel	73.1	85.2 + 9.1 (*n* = 8)	16.6
	3/7 Alg/Gel	71.7	64.2 ± 10.6 (*n* = 11)	10.5
Linear regression, complete dataset	3/4 Alg/Gel	91.0	85.2 + 9.1 (*n* = 8)	6.37
	3/7 Alg/Gel	91.0	64.2 ± 10.6 (*n* = 11)	29.5
Random forest regression, intrastudy dataset	3/4 Alg/Gel	74.0	85.2 + 9.1 (*n* = 8)	15.1
	3/7 Alg/Gel	75.3	64.2 ± 10.6 (*n* = 11)	14.7
Linear regression, intrastudy dataset	3/4 Alg/Gel	−25.9	85.2 + 9.1 (*n* = 8)	429
	3/7 Alg/Gel	−25.9	64.2 ± 10.6 (*n* = 11)	348

**Table 2 micromachines-12-00780-t002:** Predicted filament diameter values are compared against experimental values for corresponding material concentrations of alginate and gelatin. Actual values represent the mean ± standard deviation for all samples (*n* = number of samples) measured from at least three batches of Alg/Gel bioink.

Prediction Model	Material Concentration (%*w*/*v*)	Predicted Filament Diameter (μm)	Actual Value (μm)	Error (%)
Random forest regression	3/4 Alg/Gel	1037.3 μm (prediction pressure = 25 kPa)	927.6 ± 106.0 μm (*n* = 8)	10.6
	3/4 Alg/Gel	752.2 μm (predictive pressure = 103.3 kPa)	927.6 ± 106.0 μm (*n* = 8)	23.3
	3/7 Alg/Gel	857.3 μm (predictive pressure = 75 kPa)	707.2 ± 146.1 μm (*n* = 11)	17.5
	3/7 Alg/Gel	752.2 μm (prediction pressure = 103.3 kPa)	707.2 ± 146.1 μm (*n* = 11)	5.98
Linear regression	3/4 Alg/Gel	1275.8 μm (prediction pressure = 25 kPa)	927.6 ± 106.0 μm (*n* = 8)	27.3
	3/4 Alg/Gel	1149.0 μm (prediction pressure = 103.3 kPa)	927.6 ± 106.0 μm (*n* = 8)	19.3
	3/7 Alg/Gel	1187.1 μm (prediction pressure = 75 kPa)	707.2 ± 146.1 μm (*n* = 11)	40.4
	3/7 Alg/Gel	1141.3 μm (prediction pressure = 103.3 kPa)	707.2 ± 146.1 μm (*n* = 11)	38.0
Intrastudy linear regression	3/4 Alg/Gel	212.3 μm (prediction pressure = 25 kPa)	927.6 ± 106.0 μm (*n* = 8)	337
	3/7 Alg/Gel	200.0 μm (prediction pressure = 75 kPa)	707.2 ± 146.1 μm (*n* = 11)	254

**Table 3 micromachines-12-00780-t003:** Predicted extrusion pressure required to deposit material are compared against experimental values for corresponding material concentrations of alginate and gelatin. Actual values represent the mean ± standard deviation for all samples (*n* = number of batches).

Prediction Model	Material Concentration (%*w*/*v*)	Predicted Extrusion Pressure (kPa)	Actual Value (kPa)	Error (%)
Random forest regression	3/4 Alg/Gel	56.9	37.3 + 8.7 (*n* = 3)	34.4
Random forest regression	3/7 Alg/Gel	150.6	83.7 ± 4.2 (*n* = 3)	44.4
Random forest regression	8/20 Alg/Gel	150.6	208.3 ± 6.2 (*n* = 3)	38.3
Linear regression	3/4 Alg/Gel	140.8	37.3 + 8.7 (*n* = 3)	73.4
Linear regression	3/7 Alg/Gel	162.9	83.7 ± 4.2 (*n* = 3)	48.6
Linear regression	8/20 Alg/Gel	240.0	208.3 ± 6.2 (*n* = 3)	13.2

## Data Availability

The datasets that support this study are openly available in Open Science Framework (OSF) at https://osf.io/97dkx/ (last accessed on 22 June 2021).

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
