# Peer review of "Machine Assisted Experimentation of Extrusion-Based Bioprinting Systems"

_micromachines, 2021, doi:10.3390/mi12070780_

Round 1
Reviewer 1 Report
This manuscript does an excellent job on demonstrating machine learning (ML) in bioprinting parameter optimization. The data on computer science part is well presented, but data and presentation on biology part needs to be improved. I have a few concerns on the paper.
1, I would suggest authors provide more information from tissue repair and regeneration perspective. The introduction included a lot information on bioprinting limitations and rational for ML optimizing, but how tissue printing benefits tissue regenerative medicine is not provided. I think this part can potentially attract more readers. I believe this ML methods can reduce labor and expedite more accurate parameter testing in wet-lab.
2, The authors chose to use mouse neuroblastoma cells to test the methods, I am curious what is the rational to choose this specific cell line. I was hoping you can test more tissue stem cells in the future if you are interested, like airway stem cells, intestine stem cells, they can proliferate and differentiate in 3D cultures.
3, Does cell density in the "ink" influence cell viability? Have you thought about comparing high density cell ink versus low density cell ink? Or you can make more explanation on the cell density you chose.
4, Line 235, please specify the concentration of Hoechst and propidium iodide dye. Is there nutrients like growth factors in "ink" to support cell growth? That will affect cell viability.
5, Line 372, in Figure 7, representative images are blurred. Do you have high resolution image? Please show high resolution image if you still have the samples.
6, Image plate reader may not be accurate on cell viability purpose because you will loose one dimension in the samples. I would suggest confocal microscopy which will allow you acquire "Z" dimension in the images.
Reviewer 2 Report
This study develops a model that can predict cell viability based on information on concentration, solvent, polymer crosslinking information, and printing settings by applying machine-learning technology to bioprinting. I think their machine-learning techniques is not a new approach and the prediction accuracy is not very high, around 70%. However, it does show that bioprinting results can be predicted beyond a certain level by applying machine learning. Although the material system is limited to apply developed algorithms, I think it has attractive potential applications for screening bioprinting conditions. I would recommend this paper be published in its current form.
